# Measuring User Engagement with a Socially Connected, Gamified Health Promotion Mobile App

**DOI:** 10.3390/ijerph19095626

**Published:** 2022-05-05

**Authors:** Becky K. White, Sharyn K. Burns, Roslyn C. Giglia, Satvinder S. Dhaliwal, Jane A. Scott

**Affiliations:** 1School of Population Health, Curtin University, Perth 6845, Australia; becky@rhpi.com.au (B.K.W.); s.burns@curtin.edu.au (S.K.B.); 2Collaboration for Evidence, Research and Impact in Public Health, Curtin University, Perth 6845, Australia; 3Foodbank WA, Perth 6105, Australia; roslyn.giglia@foodbankwa.org.au; 4Curtin Health Innovation Research Institute, Curtin University, Perth 6845, Australia; s.dhaliwal@curtin.edu.au; 5Duke-NUS Medical School, National University of Singapore, Singapore 119077, Singapore; 6Institute for Research in Molecular Medicine (INFORMM), University of Science, Pukau Pinang 11800, Malaysia

**Keywords:** engagement, mobile health, digital health, breastfeeding, fathers, evaluation, behaviour change

## Abstract

Participant engagement is an important consideration in mHealth interventions and there are no standardised measurements available to guide researchers. This paper describes the engagement index customised for the Milk Man app, a mobile app designed to engage fathers with breastfeeding and parenting information. Participants were recruited from maternity hospitals in Perth, Western Australia. An engagement index with scores ranging from 0 to 100 was calculated. Kaplan Meier survival analysis was used to determine difference in duration of exclusive breastfeeding, and Pearson’s chi square analysis was conducted to investigate the association of engagement level with demographic characteristics and exclusive breastfeeding at 6 weeks. While overall, partners of participants who installed Milk Man were less likely to have ceased exclusive breastfeeding at any time point from birth to six weeks postpartum, this result was modest and of borderline significance (log rank test *p* = 0.052; Breslow *p* = 0.046; Tarone-Ware *p* = 0.049). The mean engagement score was 29.7% (range 1–80%), median 27.6%. Engagement level had no impact on duration of exclusive breastfeeding and demographic factors were not associated with engagement level. This research demonstrates a range of metrics that can be used to quantify participant engagement. However, more research is needed to identify ways of measuring effective engagement.

## 1. Introduction

Participant engagement in digital health promotion interventions is a vital component of intervention planning and design, and one that impacts on intervention fidelity and outcomes [1,2,3]. How people use an app, how much they use it, how long they use it for, and how much attention they pay to it, are all factors that need to be considered in terms of implementation outcomes. Engagement is often subjective and what might motivate one user to keep returning to an app might not work for another user, or might not work for them at a particular time point. There are many different aspects of mobile app development that can impact on how a person uses and keeps returning to use an intervention [2,4]. For instance, an app could be attractively designed, have the exact information a person is looking for and be perfect for their needs, yet if the app has been designed in a way whereby the loading speed between pages is too long, a user will stop engaging. 

Engagement with digital behavior change interventions (DBCI) has been defined by Perski and colleagues as: “(1) the extent (e.g., amount, frequency, duration, depth) of usage and (2) a subjective experience characterized by attention, interest and affect” [3]. Therefore, app engagement needs careful consideration of factors that increase how much people use the app, as well creating relevant, interesting and usable systems that people want to use. How best to measure engagement is a subject of current debate and research. Previous studies have used a range of methods to measure engagement including analytics only approaches [5,6], questionnaire data only approaches [7], or a combination of both [8].

Breastfeeding is the optimal first food for infants [9], however despite well-evidenced health benefits for infants and mothers [10,11], only 15% of Australian infants are exclusively breastfed to the recommended 6 months of age [12]. Fathers are influential in mothers’ decisions to initiate breastfeeding and how long they breastfeed [13,14], but can sometimes feel uncertain of their role, or not feel included in antenatal education [15]. Physical attendance at antenatal classes can be difficult due to work and other commitments, and the use of digital tools has been a recommendation from earlier research [16].

Milk Man is a gamified, socially connected mobile app designed to engage fathers in information and conversation about breastfeeding [17]. The development of the Milk Man app [18] and the description and outcomes of the randomized control trial (RCT) in which it was implemented have been described in detail elsewhere [19,20]. The app was designed by a multidisciplinary team, based on the social cognitive theory, and informed by formative research with new and expecting fathers. The app aimed to deliver time-relevant information to fathers to increase support for their partner’s breastfeeding, which we hypothesized would impact on maternal breastfeeding duration. The app contained a comprehensive information library which included evidence-based articles about a wide range of parenting topics, with a focus on breastfeeding. The app’s engagement strategies included social connectivity via a conversation forum, gamification, and push notifications (Appendix A).

In the conversation forum, fathers were placed into groups depending on their baby’s expected delivery date to enable peer connection. Twice a week, time-relevant information was delivered to users via a push notification, containing links to the conversation forum, as well as the library for further reading. Use of the app was underpinned by the gamification elements. Participants earned points, badges, and leaderboard placement for completing actions the researchers wanted to encourage, such as reading articles, commenting in the forum, or regularly checking into the app. This study aimed to define and describe participant engagement with the Milk Man mobile app, and to investigate the impact on exclusive breastfeeding at six weeks postpartum.

## 2. Materials and Methods

### 2.1. Study Overview

The Milk Man app intervention was nested in the Parent Infant Feeding Initiative (PIFI) study (ACTRN126140006056950) [19]. The PIFI was a four-armed, factorial randomised control trial aiming to test father-focussed interventions on maternal breastfeeding outcomes [20]. There was a control group, two medium intensity groups (one where fathers participated in a male-facilitated antenatal session, and one where fathers received the Milk Man app), and one high intensity group (participants received both the Milk Man app intervention and the father-focussed antenatal group). Fathers randomized to a Milk Man app group received the intervention from recruitment (antenatally, mean 32 weeks gestation) until their babies were 26 weeks of age. Questionnaires were self-completed separately by mothers and fathers at baseline (recruitment), and at six, and 26 weeks post-partum. The study received ethical approval from the Curtin University Human Research Ethics Committee (HR 82/2014; 14 May 2014) and human research ethics committees responsible for the participating hospitals.

### 2.2. Study Participants

Participants were recruited from antenatal classes conducted at maternity hospitals in Perth, Western Australia between August 2015 and December 2016. A total of 1423 couples were recruited and 681 were randomly assigned to a Milk Man app group. In order to calculate an engagement index (EI), and be included in this study, participants needed to have installed the Milk Man app and completed the six-weeks questionnaire (n = 400). Access to the app was provided at recruitment and prior to the return of the baseline questionnaire, hence some participants may have installed and evaluated the app at 6 weeks but not returned their baseline questionnaire. Completing the baseline questionnaire was not a requirement for inclusion in this analysis, and as such, demographic information was not available for some participants. In addition, to be included in the breastfeeding per-protocol analysis, the fathers needed to have matched breastfeeding data from the mother (n = 286). These data were then compared with matched participant data from the control group (n = 229). Data used in this study were collected via self-administered questionnaires from participants at baseline and again at 6 weeks postpartum. Analytics (including app usage data, library articles read, participation in the conversation forum, interaction with the gamification elements, for example) were collected directly from the customised app analytics framework [4].

### 2.3. Engagement Index

An engagement measure was developed to enable users to be grouped into differing levels of app engagement. This was informed by other measures, in particular by the Web Matrix Visitor Index [21] that was adapted to measure engagement with the Growing Healthy app used in an earlier infant feeding intervention for Australian parents [22], and the approach adopted in a gamified health promotion intervention [8]. An index was customized to suit the Milk Man intervention (See Table 1) and aimed to rank and organize participants by level of engagement. Our index used five of the seven indices proposed by the Web Matrix Visitor Index [21]. 

**Reading sub-index**: Measures the number of library articles an individual participant had viewed. This was benchmarked to the highest number read by any participant in the same time period (117 articles) to provide a percentage.**Loyalty sub-index**: Measured how often a participant had accessed the app over the time-period. App access was calculated using the number of unique days the app was opened by each participant from 30 weeks’ gestation to six weeks postpartum. This was benchmarked to the highest number of app opens by any participant, which was 62 days over the time period.**Interaction sub-index**: In Milk Man, users earned gamification points based on their interaction with app functions. A participant’s point score was used as a proxy of interaction and was benchmarked to the highest number of points scored by any participant in the time period (153 points).**Recency sub-index**: Measures how recently users returned to the app. This measure was adapted for Milk Man to measure the most recent week each participant had visited the app in weeks prior to six weeks’ postpartum (from 30 weeks gestation to 6 weeks postpartum inclusive, was 17 weeks, therefore the denominator was 17).**Feedback sub-index**: Uses information collected out of app to represent participant satisfaction. Six general statements about the participants’ overall perspectives of the app included in the 6 weeks follow-up questionnaire were used. All answers were recorded on a 5-point Likert scale, and a positive response was recorded if the participant answered strongly agree or agree to any of the questions. The sum of positive responses was divided by six to give an overall percentage.

Users were benchmarked to the highest score achieved by a participant in the cohort to give a possible range from 0–100 for each sub-index. The formulas for each sub-index are detailed in Table 1. All sub-indices were considered equal and an overall EI for each user was calculated by averaging the five sub-indices. The EI score was used to categorize participants into three equal sized groups (tertiles). These groups were then defined as poorly, moderately and highly engaged.

### 2.4. Statistical Analysis

A per-protocol analysis was conducted to determine the impact that use of the Milk Man app had on exclusive breastfeeding. A Kaplan Meier survival analysis was conducted to compare the risk of exclusive breastfeeding cessation in partners of fathers who had downloaded the app (n = 286), and those in the control group who did not have access to the app (n = 229). A Pearson chi-square test for association was conducted to investigate the association between level of engagement and demographic characteristics and exclusive breastfeeding at 6 weeks. 

## 3. Results

### 3.1. Engagement Index

The mean EI score was 29.7% (range 1–80%), median 27.6% and the standard deviation 19.8%. The overall EI scores were left skewed. Participants were divided into tertiles to define an engagement level. The range of percentages in each tertile as well as the average and number of participants is given below.

Poorly engaged (n = 133. avg. 7.94. range 1.57–17.16)Moderately engaged (n = 134. avg. 27.94. range 17.22–39.01)Highly engaged (n = 133. avg. 53.17. range 39.01–80.09).

### 3.2. Sub-Indices

Individual results of each of the sub-indices are listed below.

The average Reading sub-index score was 13.97% (std. dev. 17.84) and the median was 5.98%, which equated to seven articles read.The Loyalty sub-index scores ranged from 1.61–100% and the average was 20.50% (std. dev. 18.23). The median score was 14.52% which corresponded to the app being opened on nine unique days.The range of scores for the Interaction sub-index were 0–100% and the average was 15.42 (std. dev. 16.99). The median score of 8.82% represented a point score of 14.The median recency score was 33.33%, which translates to a last app open when their baby was 4 weeks old (calculated from 30 weeks gestation to six weeks postpartum). The mean was 49.55% (std. dev. 37.84) and range of scores were from 6.25–100%.The Feedback sub-index scores ranged from 0–100%, the average was 48.96% (std. dev. 41.41) and median score was 50%. The percentages of participants choosing strongly agree or agree for each of the individual question is listed below.
○The app was easy to use (83.4%)○The visual design was appealing (78%) ○I would recommend the app to other dads (67.2%)○The app was interesting/fun to use (60.1%)○The app made me more aware of how I can help with breastfeeding (54.6%)○The app has led to discussions with my partner. (54.1%)

### 3.3. Participant Characteristics by Engagement Level

Participant characteristics were available for 359/400 participants who had installed the app and completed the six weeks follow-up questionnaire. Two-thirds of participants had some university education and 69% identified as managers or other professionals. The majority were born in Australia and lived in the least disadvantaged areas. There was no association between engagement level and any of the participant characteristics (Table 2).

### 3.4. Duration of Exclusive Breastfeeding Per-Protocol Analysis

Compared with the control group, participants who installed Milk Man were less likely to have ceased exclusive breastfeeding at any time point from birth to six weeks postpartum (Figure 1), although this finding was of borderline significance (log rank test *p* = 0.052; Breslow *p* = 0.046; Tarone-Ware *p* = 0.049). Mean survival time for those who did not have the Milk Man app was 4.70 weeks (95% CI. 4.39–5.00) and 5.06 weeks for those who did (95% CI. 4.83–5.30). As the median (50th percentile) was not reached (Figure 1), an assessment of the difference between the means of those who downloaded the app and those who did not have access to the app is equal to 5.06–4.70 = 0.36 weeks or about 2.5 days, a positive but small difference of minor importance.

### 3.5. Duration of Exclusive Breastfeeding by Engagement Level

There was no statistically significant difference observed in the proportion of participants whose partners were exclusively breastfeeding at six weeks postpartum and any engagement level, *p* = 0.754 (Table 3).

## 4. Discussion

Our engagement index was informed by previous work [22], and customized for use in this study by benchmarking relevant sub-indices to the highest score achieved by any user. The average engagement score of 29.7% was very similar to that reported in the Growing Healthy study (mean of 30%) [22]. This approach may be useful for others seeking primarily to define and describe engagement within their own study groups, rather than comparing engagement across different studies.

The mean engagement score of 30% for both interventions may reflect the nature of breastfeeding interventions. A father whose partner is breastfeeding with no problems, may use the app much less often than one whose partner experiences difficulty. Similarly, a father whose partner ceases breastfeeding early may find little reason to stay engaged. In addition, gamification points were earnt for engagement with the conversation or polls, meaning ‘lurkers’ or those who observe but don’t interact, may have a lower score for the interaction sub-index, but may still be engaged.

Standardizing engagement measurement is complex. The level of sufficient engagement for behavior change will vary by individual. High levels of analytical driven engagement in the short-term could reflect a poorly designed app that was difficult to navigate. In addition, another study reported an association between higher engagement with the app and poorer health outcomes [23]. A smoking cessation app found those who used the *quit tips* feature of their app the most often were less likely to abstain from smoking. This was due to participants who were the heaviest smokers using the app the most.

Engagement needs to be a carefully considered component of project design and more research is needed to determine rigorous ways of reporting and understanding user engagement with digital interventions. The range of data covered by the sub-indices in the measure developed for this research will be useful to both academics and practitioners to inform both intervention design and evaluation. These considerations highlight the benefit in using a range of metrics to define engagement levels.

A per-protocol analysis involving the control group and only those couples who had downloaded the app showed promise in terms of reducing the risk of exclusive breastfeeding cessation. The effect was modest however and should be interpreted with caution. We have already reported in our primary outcomes analysis [20] that this intervention study suffered from self-selection bias in a direction to that reported for other family-based studies involving fathers [24]. In this study, most participants were highly educated and socially advantaged, both of which are positively associated with duration of exclusive breastfeeding. Thus, this self-selection bias is likely to have reduced our ability to detect a larger impact of the Milk Man app on breastfeeding outcomes.

We were unable to detect an association between level of engagement and duration of exclusive breastfeeding. Again, our ability to detect a difference between different levels of engagement with the app may have been the result of self-selection bias, and education and social advantage may be stronger predictors of breastfeeding outcomes than level of app usage. It may also mean that while we measured extent of overall engagement, we were unable to measure level of ‘effective’ engagement and looking at mechanisms to measure individual levels of effective engagement may be a more useful predictor of outcomes. This finding would benefit from further exploration in subsequent studies. More research on how to measure and report on effective engagement is needed.

This research has built on previous research in adapting and tailoring an engagement measure to fit a digital intervention. Tools for measuring engagement are still in their infancy and this study, drawing on other research, builds on the evidence. The strengths include the randomized control design and depth of information available from which to structure the engagement measure. The engagement index we described was a useful way to rank and understand participants in our own cohort.

This study had several limitations. As already described, most of our overall cohort was highly educated, lived in the least socially disadvantaged areas, were aged over 30 years and, at six-weeks postpartum, 73% were still breastfeeding—which is higher than the population average [12]. This limits the ability to generalize the findings to the general population. However, even compared with a control group exhibiting high levels of breastfeeding, the Milk Man app group showed promise in terms of positively impacting on breastfeeding outcomes.

Another limitation is that due to the need for matched breastfeeding duration data from mothers, we were unable to include all fathers who downloaded Milk Man in the engagement measure, or the breastfeeding per-protocol analysis, resulting in a reduced sample size and power of detection.

## 5. Conclusions

This study describes a tailored, adapted engagement measure that was used to describe and understand engagement with a socially connected, gamified mobile app. Even with the high level of breastfeeding overall, a modest improvement in duration of exclusive breastfeeding was observed in those with access to the Milk Man app. However, while there was a wide variation in engagement levels between participants, we were unable to detect an association between level of engagement and the primary outcome, exclusive breastfeeding at six weeks. This research demonstrates a range of metrics that can be used to quantify participant engagement but, further research is needed to identify ways of measuring effective participant engagement.

## Figures and Tables

**Figure 1 ijerph-19-05626-f001:**
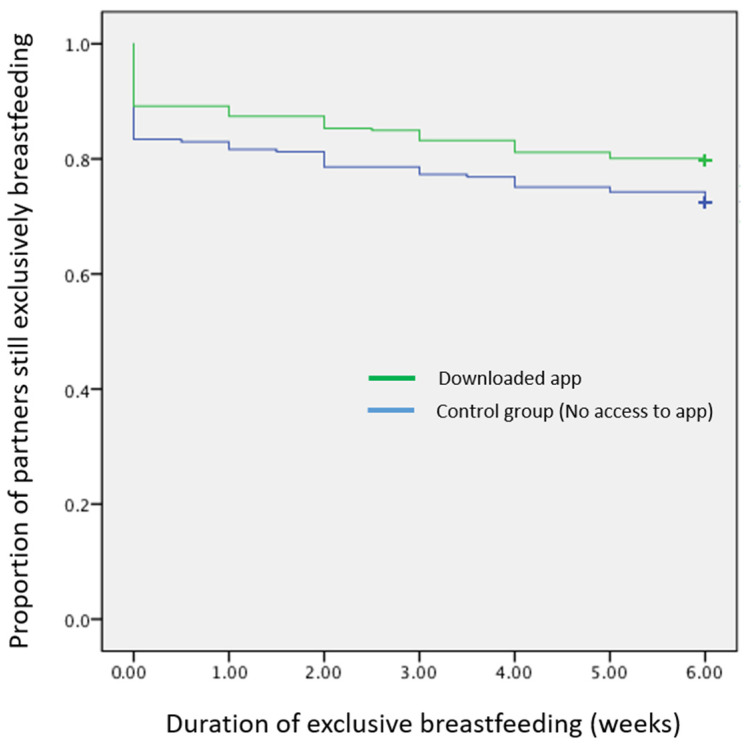
Prevalence of exclusive breastfeeding in first 6 weeks by use of Milk Man app.

**Table 1 ijerph-19-05626-t001:** Milk Man engagement index.

Sub-Indices	Description	Formula
Reading sub-index	Interaction with library contents	$=100\times \left(\frac{\mathrm{Total}otal\;library\;articles\;read}{117\;(highest\;no\;of\;articles\;read}\right)$
Loyalty sub-index	Total number of times person has accessed the app	$=100\times \left(\frac{Total\;unique\;days\;app\;was\;opened}{62\;(highest\;no\;of\;days\;app\;open}\right)$
Interaction sub-index	Gamification points percentage	$=100\times \left(\frac{Individual\;point\;score}{153\;\left(highest\;points\;score\right)\;}\right)$
Recency sub-index	Time of last visit prior to 6 weeks post-partum	$=100\times \left(\frac{1}{No\;of\;weeks\;elasped\;since\;more\;recent\;session\;\left(17-last\;week\;open\right)}\right)$
Feedback sub-index	Subjective measure, participant satisfaction with the program	$=100\times \left(\frac{No\;of\;positive\;responces\;to\;general\;MM\;questions\;\left(SA\;or\;A\right)}{All\;questions\;\left(6\right)}\right)$

**Table 2 ijerph-19-05626-t002:** Participant characteristics by engagement level.

	Engagement Level			
Participant Characteristics (n = 359)	Poor N = 133	Moderate N = 134	High N = 133	*p*-Value ^1^
Fathers age (years)	n (%)	n (%)	n (%)	
<30 (n = 62)	16 (13.9)	19 (16.0)	27 (21.6)	0.464
30–34 (n = 159)	56 (48.7)	50 (42.0)	53 (42.4)	
≥35 (n = 138)	43 (37.4)	50 (42.0)	45 (36.0)	
Father’s highest level of education				
High school/Trade (n = 118)	37 (32.5)	41 (35.0)	40 (32.0)	0.866
Some university (n = 238)	77 (67.5)	76 (65.0)	85 (68.0)	
Father’s country of birth				
Australia/New Zealand (n = 244)	80 (70.2)	73 (62.4)	91 (72.8)	0.606
United Kingdom/Ireland (n = 45)	15 (13.2)	18 (15.4)	12 (9.6)	
Africa/Middle East (n = 22)	5 (4.4)	7 (6.0)	10 (8.0)	
Asia (n = 16)	5 (4.4)	6 (5.1)	5 (4.0)	
Other (n = 29)	9 (7.9)	13 (11.1)	7 (5.6)	
Father’s occupation				
Managers and professionals (n = 246)	78 (68.4)	78 (66.7)	90 (72.6)	0.592
Other occupations (n = 109)	36 (31.6)	39 (33.3)	34 (27.4)	
IRSAD ^2^				
Deciles 1 to 6 (n = 90)	27 (23.5)	25 (21.0)	38 (30.4)	0.213
Deciles 7 to 10 (n = 269)	88 (76.5)	94 (79.0)	87 (69.6)	

^1^ Pearson chi-square *p* value ^2^ IRSAD: Index of Relative Social Advantage and Disadvantage, where 1 = most disadvantaged and 10 = least disadvantaged.

**Table 3 ijerph-19-05626-t003:** Exclusive breastfeeding at 6 weeks by engagement group.

	Engagement Level			
Exclusive Breastfeeding at 6 Weeks	Poor n (%)	Moderate n (%)	High n (%)	Total n (%)
No	24 (24.5)	30 (27.5)	27 (23.3)	81 (25.1%)
Yes	74 (75.5)	79 (72.5)	89 (76.7)	242 (74.9%)

## Data Availability

The data presented in this study are available on request from the corresponding author. The data are not publicly available due to ethical restrictions.

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
