# Peer review of "Measuring User Engagement with a Socially Connected, Gamified Health Promotion Mobile App"

_ijerph, 2022, doi:10.3390/ijerph19095626_

Round 1

Reviewer 1 Report

This paper proposes a technique for Measuring user engagement in a socially connected, gamified health promotion mobile app. An adaptive approach for a nested study. this paper has some technical flaws and need more explanations

  • First of all the title should be more precise and not like a sentence.
  • Please use the correct term in opening line of the abstract "engagement" doesn't make much sense?
  • Authors fail to explain the reason for selection of "Milk Man App". How this app generalizes medical health related issues?
  • How app engagement parameter alone improves awareness about medical related issues?
  • In the results the Download App vs Control Group has marginal improvements, how that proves efficiency of this study and proposed claim that app engagement index is the correct parameter for concluding this result.
  • Also, authors try to extend their previous work with this new results, more details needed to relate with current findings.

Author Response

This paper proposes a technique for Measuring user engagement in a socially connected, gamified health promotion mobile app. An adaptive approach for a nested study. this paper has some technical flaws and need more explanations.

Thank you for your constructive feedback which we have attempted to address.

First of all the title should be more precise and not like a sentence.

Response: We have changed the title to now read ‘Measuring user engagement in a socially connected, gamified health promotion mobile app’

Please use the correct term in opening line of the abstract "engagement" doesn't make much sense?

Response: We have changed this to now read ‘Participant engagement’.

Authors fail to explain the reason for selection of "Milk Man App". How this app generalizes medical health related issues?

Response: Thank you for this comment. We have expanded the introduction and provided a justification for the development of the Milk Man app (lines 58-61) and a more detailed description of the Milk Man app (Lines 69-79).  We have included screen shots of selected pages of the app to demonstrate the engagement strategies as supplementary material.

We are unsure what the reviewer means by ‘How this app generalizes medical health related issues’. While the app is a specific example of the use of digital technology in the growing field of mHealth, the issue of participant engagement is of general relevance to mHealth interventions, as explained in the introduction.

How app engagement parameter alone improves awareness about medical related issues?

Response: We don’t claim that the engagement parameter per see improves awareness about medical related issues. We identify participant engagement with digital technology in mHealth interventions as being crucial to the success of interventions.  In this paper we describe the customised engagement index used to measure engagement with a specific health app and investigated if the marginal overall modest effect associated with having access to the app was improved in those with a higher level of engagement.

In the results the Download App vs Control Group has marginal improvements, how that proves efficiency of this study and proposed claim that app engagement index is the correct parameter for concluding this result.

Response: The authors have not concluded that the app engagement index is the correct parameter for proving the efficacy of the study results. We’ve acknowledged that the improvement in breastfeeding was marginal and that outcome was not predicted by level of engagement. The purpose of this paper was to describe the customised app engagement index developed to measure user engagement with the Milk Man app and to investigate if higher level of engagement was associated with better outcomes compared to low levels of engagement. There remain challenges with measuring engagement which we address in the discussion, and this paper adds to the evidence in this field.

We have modified out conclusion to better reflect the purpose of this paper, which was to measure user engagement rather than on the effectiveness of the app per se (lines 301-305).

Also, authors try to extend their previous work with this new results, more details needed to relate with current findings.

Response: Reference was made to  the Engagement Analysis, conducted as part of Dr White’s PhD project, in the  published outcome evaluation of the RCT (ref 20), and the null association between level of engagement and breastfeeding outcomes acknowledged in this paper.  However, due to the complexity of the engagement analysis and the length of the RCT outcome evaluation paper, it was not possible to include the detailed engagement analysis described in this paper in the outcome evaluation paper.  Ways of measuring user engagement are still in their infancy and few papers have reported on their attempts to measure user engagement. This paper stands alone from our other work, presenting new methods and results and therefore, we believe that this paper makes an important contribution to the literature.

Reviewer 2 Report

From my perspective, the paper is super interesting with the topic that could be significant for the readers. In general, the paper is well structured and presented in proper way. However, certain parts should be improved:

I suggest adding a few details related to the app: description, features, several screens, etc. It is not clear what the app does and how it is connected with the research questions

Why WebMatrix Visitor Index was employed?

What are the implications for practicioners and academia? How the results could be used? I suggest discussion to be extended.

Author Response

From my perspective, the paper is super interesting with the topic that could be significant for the readers. In general, the paper is well structured and presented in proper way. However, certain parts should be improved:

Thank you for your comments and constructive review

I suggest adding a few details related to the app: description, features, several screens, etc. It is not clear what the app does and how it is connected with the research questions

We have expanded the introduction and provided a justification for the development of the Milk Man app (lines 58-61) and a more detailed description of the Milk Man app (Lines 69-79).  We have included screen shots of selected pages of the app to demonstrate the engagement strategies as supplementary material.

Why WebMatrix Visitor Index was employed?

Response: The WebMatrix Index was adapted for use by another Australian infant feeding intervention, the Growing Healthy study (Line 121-122). Ways of measuring engagement are in their infancy and our research aimed to build on the available evidence and tools, which at the time was limited and included the WebMatrix Index and the Growing Healthy study.

What are the implications for practitioners and academia? How the results could be used? I suggest discussion to be extended.

Response: Thank you for this comment. We have added to the discussion section. ‘Engagement needs to be a carefully considered component of project design and more research is needed to determine rigorous ways of reporting and understanding user engagement with digital interventions. The range of data covered by the sub-indices in the measure developed for this research will be useful to both academics and practitioners to inform both intervention design and evaluation. These considerations highlight the benefit in using a range of metrics to define engagement levels.’

Reviewer 3 Report

Paper entitled "Measuring user engagement in a socially connected, gamified
health promotion mobile app. An adaptive approach for a nested study"  describes a tailored, adapted engagement measure that was used to describe and understand engagement with a socially connected, gamified mobile app. The Paper is well written with sufficient evidence of its finding on the current topic. 

"In the study there 286 was a wide variation in engagement levels between participants, however, engagement 287 level did not predict breastfeeding outcomes, and nor did any demographic factors predict 288 engagement level." -----If you could explain this sentence better way it will be great. 

Author Response

Paper entitled "Measuring user engagement in a socially connected, gamified
health promotion mobile app. An adaptive approach for a nested study"  describes a tailored, adapted engagement measure that was used to describe and understand engagement with a socially connected, gamified mobile app. The Paper is well written with sufficient evidence of its finding on the current topic. 

Thank you for your comments and constructive review

"In the study there  was a wide variation in engagement levels between participants, however, engagement level did not predict breastfeeding outcomes, and nor did any demographic factors predict engagement level." -----If you could explain this sentence better way it will be great. 

Response: We have modified out conclusion to better reflect the purpose of this paper, which was to measure user engagement rather than on the effectiveness of the app per se (lines 301-305), and this sentence is no longer in the conclusion.

Reviewer 4 Report

The authors present a model for measuring user participation in a mobile application without clearly showing the MilK Man App. It is necessary that the authors show the design of the App, interface, usability, menu, etc.

Without knowing the Milk Man App, it is difficult to understand if it is a gamified and socially connected health promotion app.

In the Introduction section it is necessary to answer What are the most common metrics and KPIs of participation in mobile applications?

The study presented by the authors shows a personalized engagement index for the Milk Man app, but in reality it is a strategy to measure the engagement of this app. But, what is the optimal experience for the user?

Strongly agree with the authors who confirm that it is a biased sampling (self-selection). This can bias the discussion of results and logically the conclusions. The sample needs to be more diverse.

Finally, it is necessary to detail the adaptive approach for a nested study. In the manuscript this approach should be discussed.

Please clarify this date: The study received ethical approval from the Curtin University Human Research Ethics Committee (HR 82/2014; May 14, 2014) and human research ethics committees responsible for the participating hospitals.

Author Response

The authors present a model for measuring user participation in a mobile application without clearly showing the Milk Man App. It is necessary that the authors show the design of the App, interface, usability, menu, etc. Without knowing the Milk Man App, it is difficult to understand if it is a gamified and socially connected health promotion app.

Response: Thank you for your constructive feedback. We have expanded the introduction and provided a justification for the development of the Milk Man app (lines 58-61) and a more detailed description of the Milk Man app (Lines 69-79).  We have included screen shots of selected pages of the app to demonstrate the engagement strategies as supplementary material.

In the Introduction section it is necessary to answer What are the most common metrics and KPIs of participation in mobile applications?

Response: Due to the differences in mobile app design, there are no set KPIs. Common metrics include those described in lines 35-36. The extent of use – ie how much people use it, how long they spend using it, as well as how much they enjoy using it.

The study presented by the authors shows a personalized engagement index for the Milk Man app, but in reality it is a strategy to measure the engagement of this app. But, what is the optimal experience for the user?

Response: This is a very good question, and one which we touch on in discussion where we say ‘It may also mean that while we measured extent of overall engagement, we were unable to measure level of ‘effective’ engagement and looking at mechanisms to measure individual levels of effective engagement may be a more useful predictor of outcomes. This finding would benefit from further exploration in subsequent studies. More research on how to measure and report on effective engagement is needed.’

We agree these questions are important and more research into individualised levels of optimal engagement is needed. As our study did not report any significant findings with engagement level and breastfeeding outcomes, we are unable to say what the optimal level of engagement for breastfeeding outcomes may be.

Strongly agree with the authors who confirm that it is a biased sampling (self-selection). This can bias the discussion of results and logically the conclusions. The sample needs to be more diverse.

No response required

Finally, it is necessary to detail the adaptive approach for a nested study. In the manuscript this approach should be discussed.

Response: We have changed the title to now read ‘Measuring user engagement in a socially connected, gamified health promotion mobile app’.  We were referring to the adaption of the Web Matric Visitor Index for use in this study which was an analysis nested in the evaluation of the overarching Parent Infant Feeding Initiative RCT.

Please clarify this date: The study received ethical approval from the Curtin University Human Research Ethics Committee (HR 82/2014; May 14, 2014) and human research ethics committees responsible for the participating hospitals.

Response: We can confirm that ethical approval was obtained in May 2014 as indicated, This included approval not only for the PIFI RCT but also approval to conduct the formative research (e.g. focus groups and beta-testing) with end users during the development of the Milk Man app, which took approximately 12 months and is described in reference 18.  The PIFI RCT began with recruitment which was conducted between August 2015 and December 2016 with the last recruited participants  finishing the intervention in June 2017. The development of the Engagement Index formed part of Becky Whites PhD thesis which was completed in 2018. Due to the complexity of the paper, this was not included in the final paper reporting the outcomes of the PIFI intervention published in reference 20.  While at first glance the results may seem dated, due to continued lack of reporting of methodology related to the measurement of user engagement with mHealth intervention we believe that this paper makes an important contribution to the literature.

Round 2

Reviewer 1 Report

The authors have addressed my concerns and provided enough details. The paper can be accepted after minor spell check and revision.

Reviewer 4 Report

The authors have reviewed and corrected all comments and suggestions made.
Thank you